

# Building heat-resilient Caribbean reefs: integrating thermal thresholds and coral colonies selection in restoration

Macarena Blanco Pimentel[1,2], Johanna Calle-Triviño[3], Daniel J. Barshis[4], Sancia E.T. van der Meij[2,5] and Megan K. Morikawa[6]

[1] Wave of Change, Iberostar Group, Bavaro, Dominican Republic
[2] Groningen Institute for Evolutionary Life Sciences (GELIFES), University of Groningen, Groningen, Netherlands
[3] Wave of Change, Iberostar Group, Quintana Roo, Mexico
[4] Department of Biological Sciences, Old Dominion University, Norfolk, VA, United States of America
[5] Naturalis Biodiversity Center, Leiden, Netherlands
[6] Salesforce, Washington, United States of America

Corresponding author
Macarena Blanco Pimentel,
m.blanco.pimentel@rug.nl

## ABSTRACT

Caribbean reefs face increasingly frequent and intense bleaching events, adding to the numerous other threats impacting these ecosystems. Addressing these challenges requires global action to reduce climate drivers, along with local efforts like reef restoration. Active restoration using thermotolerant coral colonies offers a potential strategy to alleviate these impacts; however, gaps remain in identifying context-specific temperature thresholds to guide colony selection and standardize thermotolerance assessment methods. This study addressed these gaps in two phases. First, by determining practical thresholds to differentiate species responses to heat stress; and second, by developing a framework to identify and prioritize resilient colonies for restoration. In the first phase, 70 colonies of *Acropora cervicornis*, *Diploria labyrinthiformis*, *Montastraea cavernosa*, *Orbicella annularis*, *O. faveolata*, *Porites astreoides*, and *P. porites* were sampled from reefs in the southeastern Dominican Republic. Heat stress responses were assessed through 3-hour heat pulse assays above the local maximum monthly mean (MMM) temperature, combining visual bleaching ranks, pixel intensity as a proxy for chlorophyll loss, and pulse amplitude modulated (PAM) fluorometry. Species-specific $T_{50}$ thresholds were identified as the temperatures where 50% of colonies showed signs of stress. In the second phase, intraspecific thermotolerance was further examined for *D. labyrinthiformis*, *M. cavernosa*, *O. annularis*, *O. faveolata*, and *P. astreoides* using 99 colonies from known parent sources. Heat pulse assays at control (MMM) and $T_{50}$ temperatures were repeated four times to assign colony-specific thermal performance scores. This study integrates inter- and intraspecific thermotolerance data into a practical selection framework, offering valuable insights to guide restoration under climate change.

## INTRODUCTION

The beginning of reef degradation in the Caribbean dates to the 1950s, primarily driven by overfishing and the increasing proximity of human settlements (*Pandolfi et al., 2003*; *Cramer et al., 2020*). Since then, it has accelerated due to continued overfishing, disease outbreaks, thermal stress, sedimentation, nutrient enrichment, hurricane impacts and chemical pollution (*Jackson et al., 2014*; *Precht et al., 2020*; *Cramer et al., 2021*). The most drastic recent losses in coral cover and diversity in the Caribbean are attributed to disease outbreaks (*Precht et al., 2016*), marine heatwaves and subsequent bleaching (*Eakin et al., 2010*; *Hoegh-Guldberg et al., 2023*). Furthermore, the relationship between ocean warming and disease outbreaks seems to be related, exacerbating the severity of the threats in the region (*Cróquer & Weil, 2009*; *Randall & Van Woesik, 2015*).

To address these challenges, it is essential to take measures to reduce anthropogenic carbon emissions and unsustainable human development. Alongside these efforts, reef restoration has emerged as a potential strategy for improving the health and functionality of these ecosystems. However, the restoration process is complex, involving a variety of organisms and ecological interactions. To be effective, restoration should also account for persistent global stressors such as climate change, highlighting the need for continued research into best practices to improve long-term outcomes. As a result, it is important to approach restoration with careful planning, focusing on the resilience of the reefs and the long-term sustainability of recovery efforts (*Hughes et al., 2023*).

One component of these strategies aims to reduce bleaching in coral nurseries and restoration sites by incorporating locally selected coral individuals that may be more tolerant to higher temperatures, as differences in heat stress tolerance are evident not only between species but also among individuals of the same species (*Caruso, Hughes & Drury, 2021*; *Cunning et al., 2021*; *Klepac et al., 2024*). While bleaching does not always result in mortality and may not fully predict long-term performance of surviving corals (*e.g.*, growth or reproduction), selecting thermotolerant phenotypes may still increase the chances of persistence and general ecosystem resilience (*Morikawa & Palumbi, 2019*).

Thermal tolerance in corals is influenced by a complex interplay of factors, many of which remain poorly understood (*Suggett & Smith, 2020*). Advances in approaches such as gene expression, proteomics and metabolomics continue to deepen our understanding of the mechanisms underpinning heat stress responses (*Kirk et al., 2018*; *Fuller et al., 2020*; *Roach et al., 2021*). While these methods offer detailed insights, other techniques can provide a more immediate and general overview of a coral's physiological stress state and the status of its symbiotic relationship with algal partners. For instance, pulse amplitude modulated (PAM) chlorophyll fluorometry is a widely used tool to assess photosynthetic efficiency and symbiont health (*Kitajima & Butler, 1975*; *Maxwell & Johnson, 2000*), and it has been demonstrated to be useful in identifying potentially thermo-tolerant individuals for restoration efforts (*Cunning et al., 2021*). Complementary methods such as pixel intensity analysis and visual bleaching scores offer additional layers of information by capturing cumulative changes in pigmentation or symbiont density. These metrics,

though sometimes more subjective, may reflect stress responses that are not captured by instantaneous physiological proxies like photosynthetic efficiency.

The selection of potentially thermotolerant individuals can be carried out following different approaches as described by *Caruso, Hughes & Drury (2021)*. Stress assays, for example, simulate conditions akin to natural bleaching events, allowing the acceleration of the selection process that would occur naturally. However, this procedure presents several challenges. Key questions include determining the optimal stress evaluation method (*Nielsen et al., 2022*) and addressing comparability of results obtained through different temperature systems (*Grottoli et al., 2021*; *Evensen et al., 2024*). Moreover, the extent to which laboratory temperature profiles accurately reflect natural conditions remains unclear (*Voolstra et al., 2020*), as well as the ideal season for conducting stress assays (*Chapron et al., 2022*), and the optimal number of coral colonies and collection sites needed to ensure variability in responses (*Grottoli et al., 2021*).

Additionally, the selection of thermotolerant individuals through heat stress experiments requires identifying a temperature that elicits a variable bleaching response across individuals of a given species since too low or too high a temperature can obscure variability, and this temperature may vary between species (*Morikawa & Palumbi, 2019*; *Cunning et al., 2021*). Although dose–response approaches such as $ED_{50}$ are commonly used to define bleaching thresholds, particularly in coral bleaching automated stress system (CBASS) assays, which have become widely adopted and validated (*Voolstra et al., 2020*; *Cunning et al., 2024*), our study adopts a complementary approach. Specifically, we derive median threshold temperatures ($T_{50}$) based on variability across multiple stress metrics, including photophysiological and visual indicators. This strategy supports higher replication per genotype and facilitates the identification of thermotolerant individuals under short-term, restoration-relevant conditions. By building on the principles of CBASS, this method offers an alternative framework tailored to practical restoration applications, while acknowledging that the predictive power of such short-term assays ultimately depends on long-term monitoring of outplanted coral performance.

This study focuses on advancing coral reef restoration by identifying thermotolerant coral colonies through a standardized, ecologically relevant heat stress assay. Specifically, it aims to determine the temperatures that elicit the greatest variability in stress responses among seven Caribbean stony coral species, helping to define context-specific thresholds useful for both interspecific comparisons and restoration planning. The first objective addresses the interspecific bleaching response, asking: how do coral species differ in their heat stress responses and bleaching thresholds across varying temperatures and evaluation methods? Building on these results, the second objective provides a practical framework for identifying and prioritizing colonies with naturally higher thermotolerance, using species-specific thresholds derived from the first phase. This is essential for understanding intraspecific variation and guiding colony selection in restoration programs. Together, these objectives respond to the need for robust yet adaptable selection tools by integrating multiple stress proxies and realistic temperature exposures.

Ultimately, the study addresses key knowledge gaps in Caribbean coral bleaching thresholds (*McLachlan et al., 2020*) and the identification of resilient genotypes for

restoration (*Klepac et al., 2024*), recognizing that such thresholds are shaped by local biological and environmental conditions. It also provides practical guidance on appropriate sampling effort to capture response variability and proposes a replicable framework for comparing thermotolerance across taxa and sites. By employing a standardized yet flexible approach, this work offers valuable insights and tools for conservation initiatives seeking to enhance coral resilience in the face of climate change.

## MATERIALS & METHODS

### Study overview

The study took place from 2021 to 2022 in the Dominican Republic (DR). Coral samples were obtained from three reefs in the southeastern region (Appendix S1: Fig. S1) (Coco Reef: 18.33668N, 68.8239W; Playita: 18.3728N, 68.8532W; and Magayan: 18.3609N, 68.8453W). Coco Reef and Magayan have similar maximum depths of 12 to 13 m, while Playita is a shallower reef with a maximum depth of eight m. The study targeted seven coral species that represent primary reef builders in the Caribbean and represent different life-history strategies (*Darling et al., 2012*) and general morphologies: *Acropora cervicornis* (Lamarck, 1816) (branching), *Diploria labyrinthiformis* (Linnaeus, 1758) (massive), *Orbicella annularis* (Ellis & Solander, 1786) (columnar), *O. faveolata* (Ellis & Solander, 1786) (boulder), *Montastraea cavernosa* (Linnaeus, 1767)(mounding), *Porites astreoides* (Lamarck, 1816) (submassive) and *P. porites* (Pallas, 1766) (digitate). Each species was studied separately, and specific experiment dates can be found in Appendix S1: Table S1.

Corals were collected under required permits from the Ministry of the Environment of the Dominican Republic (DJ-C-1-2021-00155; VRCyM-0226-2022). For each species, the study was divided into two experimental phases. In the first phase, the interspecific heat stress response was evaluated, and bleaching thresholds ($T_{50}$) were identified as the temperature at which stress responses reached around 50% of the maximum values observed at the highest temperature treatment. This 50% point was estimated using different criteria for each metric, aiming to select a temperature that showed intermediate or varied responses (*Morikawa & Palumbi, 2019*; *Evensen et al., 2021*). In the second phase, multiple colonies of each species were exposed to their species-specific $T_{50}$ to evaluate intraspecific variability and identify the most thermotolerant individuals.

The experimental design was informed by the CBASS framework (*Voolstra et al., 2020*; *Evensen et al., 2021*), particularly in the use of standardized short-term thermal ramping profiles. A custom-built temperature control system developed at our laboratory was used to replicate these conditions with high precision across multiple tanks, offering stable and consistent thermal regulation adapted to our setup. Experiments were conducted at the Coral Lab of the Iberostar Hotel (Bavaro, DR) using four 120 L acrylic tanks with respective sump tanks that run as an open water system. Seawater was pumped from a 98 m deep saltwater well and water was constantly renewed at minimum rate (1 L/min). The temperature control system consisted of a locally designed automated programmable logic controller (PLC) coupled with submerged temperature sensors and solenoid valves to regulate fresh cold/warm water inlet opening to an exchanger connected to the sump

tank, from where the water was constantly recirculated between the main tank at a rate of 10 lpm. Cold and warm water was stored in two tanks connected to a primary cooling circuit through a chiller (ECO-1 1/2HP, Ecoplus) and a heating circuit through a heater (CSPAXI55, Hayward, CA, USA), and to a secondary circuit that reached the exchangers. Cold water was kept at 10 °C and warm water at 45 °C. An interface software (VNC viewer) allowed setting temperature profiles independently for each tank with a performance precision of +/- 0.1 °C. Lighting was provided using LED aquarium lights (Hydra 52HD, AquaIllumination, Bethlehem, PA, USA) set to a PAR of ∼250 µmol quanta $m^{-2}$ $s^{-1}$ (SQ-420, Apogee, Logan, UT, USA) in a 12:12 h cycle with a 3-hour ramp up and down to mimic sunrise and sunset.

## Interspecific response (Phase 1)
### Experimental design
For each species, samples from 10 colonies were collected across the three reefs except for *A. cervicornis*, which were collected from the rope structures of the Iberostar in-water nursery located at Coco Reef. Wild colonies were randomly selected, ensuring they were apparently healthy, that the samples collected represented no more than 10% of each colony, and that a minimum distance of five m was maintained between them (*Baums et al., 2019*). For *A. cervicornis*, only the apical tips of the ramets were used as experimental replicates, while for *P. porites*, branches were cut longitudinally and all resulting sections were used. For the remaining species, fragments were collected from the skirt or lower lateral portions of the colonies, where tissue was more accessible. Samples were kept in outdoor tanks (raceways) of the coral laboratory for the night before the start of the experiments. Raceways received water from the same open system as experimental tanks. Each colony was divided using a diamond band saw (Gryphon) into eight fragments (of around five $cm^2$), one per temperature treatment. However, in this phase, different colonies were used as replicates for each temperature, with each temperature treatment having 10 replicates per species. Temperature treatments included one control temperature that corresponded to the maximum monthly mean (MMM) temperature of the area (28.5 °C) (NOAA Coral Reef Watch database) and seven temperatures that ranged from +3 to +9 °C over the MMM. However, as only four tanks could be run at a time, samples were divided into two sets, each of which was used for four temperature treatments over two consecutive days (Fig. 1). For all treatments, temperature ramps started at noon and continued for the following 3 h rising from 28.5 °C to the maximum temperature set for each tank. Maximum temperature was held for 3 h before being gradually reduced to the control value over the course of one hour. This temperature ramping scheme followed that of other experiments aimed at assessing coral bleaching thresholds and physiological responses (*Voolstra et al., 2020*; *Evensen et al., 2021*).

### Heat stress evaluation
Heat stress response was assessed by three methods. The first method measured photosynthetic efficiency ($F_v/F_m$) through pulse amplitude modulated (PAM) chlorophyll fluorometry (Junior-PAM, Walz) in dark-acclimated samples (*Kitajima & Butler, 1975*; *Maxwell & Johnson, 2000*). Dark acclimation period corresponded to the 30 min following

**Figure 1** **Experimental design of both inter- and intra-specific experimental phases.** Species were studied independently following this design. MMM corresponds to the maximum monthly mean temperature of the area (28.5 °C); $T_{50}$ corresponds to the temperature that elicited 50% of the stress response. PAM refers to pulse amplitude modulated chlorophyll fluorometry. BL refers to bleaching response evaluation.

the end of the light cycle which also corresponded to the last 30 min of the temperature ramp (Fig. 1). PAM measurements were taken at this point. $F_v/F_m$ quantifies physiological stress due to photoinhibition of symbionts at a specific moment in time, providing a snapshot of photosynthetic function under heat stress conditions. This metric is considered one of the most quantitative approaches to assess thermal stress in corals and is often used as a comparative standard in stress assays (*Cunning et al., 2021*; *Klepac et al., 2024*).

The other two methods evaluated color changes as a proxy of symbiont loss during bleaching: visual scoring of the fragments (modified from *Edmunds, Gates & Gleason, 2003*; *Siebeck et al., 2006*) and photographic analysis of chlorophyll content through pixel intensity (*Winters et al., 2009*). These metrics reflect cumulative bleaching effects throughout the experimental period, which may result from the loss of symbionts, their chlorophyll pigments, or both (*Fitt et al., 2001*). Bleaching assessments took place the morning after each experimental day, before the end of an experimental cycle. This timing was selected because visible signs of bleaching often develop gradually and become more apparent overnight, as physiological damage progresses after heat exposure (*Fitt et al., 2001*; *Lesser, 2021*). Fragments were visually evaluated and assigned a bleaching score from 1 (non-bleached) to 5 (totally bleached). Scores 2, 3, and 4 corresponded to "visually", "moderately" and "severely" bleached categories, respectively (see Appendix S2: Fig. S2 for reference images of each species and corresponding bleaching score category). To ensure consistency and minimize observer bias, all visual assessments were conducted by the same team member. Fragments were subsequently photographed next to a grayscale reference (Kodak Color Separation Guide and Gray Scale Q-13) for later image calibration and analysis using MATLAB R2019a version 9.6.0.1072779 (The MathWorks Inc., Natick, MA, USA). Image analysis was based on the quantification of pixel intensities from red, green,

and blue channels. Red channel pixel intensity was used as a proxy for loss of chlorophyll density as a result of bleaching (*Winters et al., 2009*), where an increase in the pixel intensity within the red channel would be proportional to an increase in tissue whitening associated with environmentally induced stress (*Voolstra et al., 2020*).

This multi-metric approach was chosen to provide complementary perspectives on thermal stress responses, capturing both real-time physiological effects and cumulative bleaching outcomes. The inclusion of a rapid visual score, though subjective, allowed for field-applicable insights and provided immediate feedback on the variability of bleaching expression across samples. Equal weight was assigned to the three stress evaluation methods, as each captures distinct and complementary aspects of the bleaching response, enhancing the robustness and applicability of genotype selection under restoration-relevant conditions.

### $T_{50}$ determination

The objective of this first experimental phase was to find the $T_{50}$ temperature for each species. This temperature may vary depending on the stress assessment method used and therefore a $T_{50}$ temperature was obtained for each of the three methods in this study. To select the final $T_{50}$ temperature for the second intraspecific phase, the mean of the $T_{50}$ of the three methods was calculated and rounded according to the results obtained to ensure that the temperature chosen would elicit variable responses across methods. For both photosynthetic efficiency and pixel intensity, data from the ten colonies were first averaged by treatment. The temperature treatment corresponding to the median of these two data frames was selected as $T_{50}$. If the median value corresponded to the value between two temperatures, the intermediate temperature was selected. Regarding bleaching visual rank, the $T_{50}$ temperature corresponded to the treatment that elicited a more variable response among the colonies, resulting in a greater number of different bleaching scores or outcomes. There were some exceptions to this rule in which the temperature in between two treatments was selected. In cases where two adjacent temperatures yielded similar distributions of bleaching scores, or where a treatment showed >50% of fragments at the extremes (*e.g.*, fully bleached or not bleached), intermediate temperatures were selected to better capture the range of variation. This decision rule was applied consistently across species and aimed to identify a temperature that maximized the ability to discriminate among individual responses without inducing uniformly severe or negligible bleaching. Given the applied objective of selecting thermotolerant colonies under short-term stress exposure, this approach prioritized ecological relevance and response variability over rigid parametric thresholds.

## Intraspecific response (Phase 2)
### Experimental design

In this phase, five of the seven species from phase 1 were studied: *D. labyrinthiformis*, *O. annularis*, *O. faveolata*, *M. cavernosa*, and *P. astreoides*. For each species, a new set of samples from 19 to 20 colonies were collected across the three reefs. This time, colonies were not randomly collected but selected from Iberostar's wild parent colonies repository, which is composed of colonies that have been previously labeled, mapped, photographed,

and are periodically monitored. Sample collection and fragmentation were conducted as described in Phase 1. Following the same protocol, samples were kept in outdoor raceways, and experiments began a day after collection. Each colony's tissue was divided into eight fragments, and assays were carried out in four tanks over two consecutive days. However, only two temperature treatments were used: control (MMM, 28.5 °C) and $T_{50}$ temperature (variable depending on the species) (Fig. 1). This way, four replicates per treatment and colony were employed. This single-temperature approach, based on the species-specific $T_{50}$ identified in Phase 1, was chosen to prioritize high replication across colonies and align with restoration-relevant applications, rather than conducting full dose–response assays as in the CBASS framework. Both temperature profiles and heat stress evaluation methods were performed following the same protocol as in the interspecific phase.

### *Ranking of colonies*

Each colony was assigned a thermotolerance score from 1 (most thermotolerant) to 4 (least thermotolerant) based on the three heat stress evaluation methods. Due to the distinct nature of these methods, different scoring procedures were applied, summarized below and explained in detail in Appendix S2.

For photosynthetic efficiency ($F_v/F_m$), only heated samples were used for thermotolerance scoring, while control samples ensured no additional stress factors. $F_v/F_m$ average values were calculated from heated replicates for each colony. Thermotolerance scores were assigned based on established stress ranges (*Evensen et al., 2022*): $0.5 \geq$ value $\geq 0.7$: Score 1; $0.4 \geq$ value $> 0.5$: Score 2; $0.3 \geq$ value $> 0.4$: Score 3; value $< 0.3$: Score 4. For pixel intensity, control values were used to standardize heated values due to variability in coloration across colonies. This way, average values were calculated for both control and heated treatments, and a ratio was calculated between the heated and control averages; higher ratios indicated greater bleaching. These ratios were divided into quartiles (Q) for thermotolerance scoring: value $\leq$ Q1: Score 1; Q1 $<$ value $\leq$ Q2: Score 2; Q2 $<$ value $\leq$ Q3: Score 3; Q3 $<$ value $\leq$ Q4: Score 4. For bleaching visual rank, bleaching scores from the four heated replicates were summed for each colony, resulting in totals ranging from 4 (1+1+1+1) to 20 (5+5+5+5). Colonies received thermotolerance scores as follows: 4–8: Score 1 (most thermotolerant); 9–12: Score 2; 13–16: Score 3; 17–20: Score 4 (least thermotolerant). After assigning scores for each method, the scores were summed for each colony, and colonies were ranked from most thermotolerant (lowest total score) to least thermotolerant (highest total score).

It is worth noting that, unlike $F_v/F_m$ and visual bleaching scores, which have recognized stress thresholds, pixel intensity lacks standardized reference values to interpret bleaching severity. For this reason, we calculated relative color retention (compared to control colonies) and used a quartile-based approach to assign thermal scores. While this method ensures a full distribution of scores (1 to 4), it may inherently introduce more variability compared to the other metrics.

## Statistical analyses

Statistical analyses and graphics were performed using R version 4.2.1 (*R Core Team, 2022*) within RStudio version 1.4.1106 (RStudio Team, Boston, MA, USA). Given the
non-normally distributed data, a Kruskal–Wallis test was employed to assess differences in heat stress responses among the three methods (photosynthetic efficiency, pixel intensity, and bleaching visual rank) across species at all temperatures and at each temperature (Phase 1). A Nemenyi test was used for *post-hoc* comparisons between species and temperatures, and a principal component analysis (PCA) was performed to evaluate the spread of data on heat stress across the three methods. To conduct this analysis, each subject corresponded to one species under a specific temperature treatment. To test whether the distinct groupings observed in the multivariate stress response data represented statistically significant differences, we first added a categorical variable to the dataset corresponding to performance groups visually identified from the PCA. A PERMANOVA was then conducted using the *adonis2()* function from the vegan package in R, based on Euclidean distances among standardized values of the three stress metrics ($F_v/F_m$, pixel intensity, and visual bleaching scores), with 999 permutations. Pixel intensity and bleaching visual values were inverted, as lower values indicate better thermotolerance performance for these methods, in contrast to photosynthetic efficiency, where higher values are preferred. Spearman's rank correlation was also used to corroborate the correlation between factors. For intraspecific results (Phase 2), a PCA was carried out using all species and colonies data across the three methods. Loadings from the first principal component (PC1) were used to obtain individual scores that were then compared with the total thermotolerance score given to each subject following the scoring procedure of this study to explore correlation. The correlation was tested using a Spearman's rank statistic.

## RESULTS

### Interspecific response (Phase 1)
#### *Photosynthetic efficiency*
Photosynthetic efficiency ($F_v/F_m$) demonstrated non-stressed optimal values (0.5–0.6) (*Evensen et al., 2022*) for samples at lower temperatures, and a decline in values became apparent across all species after reaching approximately 35 °C (Fig. 2A). At 36.5 °C (MMM+8), *A. cervicornis*, *M. cavernosa*, and *Porites* spp. still had high $F_v/F_m$ mean values of around 0.4. However, at the highest temperature (37.5 °C; MMM+9), only *A. cervicornis*, and *P. porites* mantained high $F_v/F_m$ mean values. When comparing the photosynthetic efficiency between species across all temperatures, significant differences were found (Kruskal–Wallis; $\chi^2$ (6) = 88.971, $p < 2.2e{-}16$), especially driven by *A. cervicornis* and *P. porites*, which differed from all other species but not from each other (Appendix S1: Table S2), as both species maintained $F_v/F_m$ values above 0.3 at high temperatures. Further analysis of species differences at each temperature revealed significant effects across all tested temperatures, with most pairwise differences involving *A. cervicornis* or *P. porites* (Appendix S1: Table S3). The temperatures at which the highest number of interspecific differences were observed were 35.5 °C (MMM+7) and 36.5 °C (MMM+8). Median values of $F_v/F_m$ (Appendix S1: Table S4) revealed the highest temperature thresholds for *M. cavernosa*, followed by *A. cervicornis*, *O. faveolata*, and *P. porites*; whereas the lowest threshold corresponded to *P. astreoides* (Table 1).

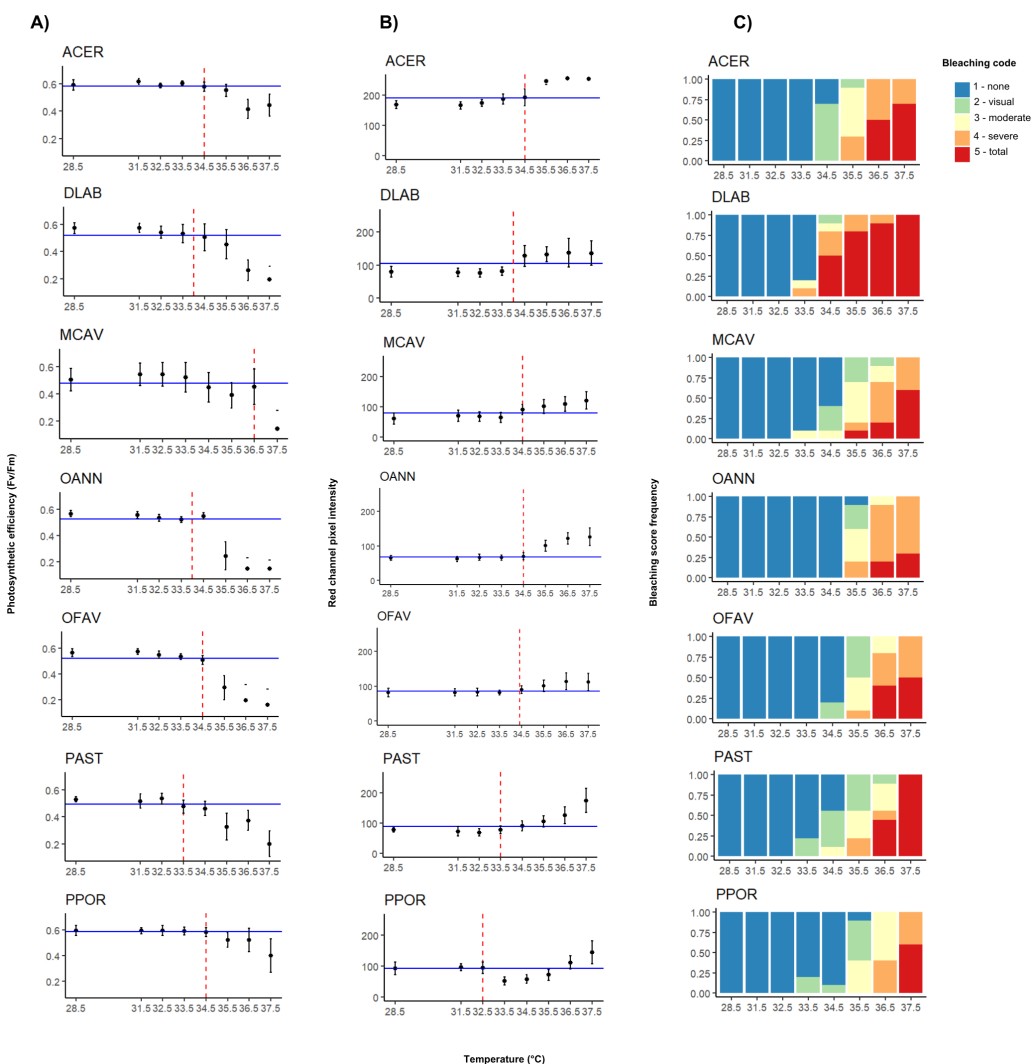

**Figure 2** Interspecific heat stress response results of photosynthetic efficiency ($F_v/F_m$) (A), pixel intensity analysis (B), and bleaching visual rank (C). ACER, *Acropora cervicornis*; DLAB, *Diploria labyrinthiformis*; MCAV, *Montastraea cavernosa*; OANN, *Orbicella annularis*; OFAV, *Orbicella faveolata*; PAST, *Porites astreoides*; PPOR, *Porites porites*. The blue line represents the median value of the data set for both photosynthetic efficiency and pixel intensity. The red dashed line represents the temperature value extrapolated from the median, *i.e.,* the $T_{50}$ temperature.

## *Pixel intensity analysis*

Red channel pixel intensity differed significantly between species across temperatures, (Kruskal–Wallis; $\chi^2$ (6) = 209.96, $p < 2.2e{-}16$), with the highest values consistently observed in *A. cervicornis*, which differed from all other species (Appendix S1: Table S5). As shown in Fig. 2B, *A. cervicornis* exhibited notably brighter coloration (higher pixel intensity values), in contrast to species such as *M. cavernosa*, *Orbicella* spp., and *P. porites*, which consistently displayed lower intensities across the temperature range. Significant differences were also found between *D. labyrinthiformis* and both *O. annularis* and *M. cavernosa* at

**Table 1** $T_{50}$ temperature values for each species across methods and the final $T_{50}$ selected for the second experimental phase.

| Species | Relative $T_{50}$ temperature values (°C above MMM) | | | Absolut $T_{50}$ temperature values (°C) | | | Averaged $T_{50}$ | Selected $T_{50}$ |
|---|---|---|---|---|---|---|---|---|
| | $F_v/F_m$ | Pixel intensity | Bleaching visual | $F_v/F_m$ | Pixel intensity | Bleaching visual | | |
| ACER | MMM+6 | MMM+6 | MMM+7 | 34.5 | 34.5 | 35.5 | 34.8 | NA |
| DLAB | MMM+5.5 | MMM+5.5 | MMM+5.5 | 34 | 34 | 34 | 34 | 34 |
| MCAV | MMM+8 | MMM+6 | MMM+7.5 | 36.5 | 34.5 | 36 | 35.7 | 36 |
| OANN | MMM+5.5 | MMM+5.5 | MMM+7 | 34 | 34.5 | 35.5 | 34.7 | 35.5[*] |
| OFAV | MMM+6 | MMM+5.5 | MMM+7.5 | 34.5 | 34.5 | 36 | 35 | 35.5[*] |
| PAST | MMM+5 | MMM+5 | MMM+7.5 | 33.5 | 33.5 | 36 | 34.3 | 35.5[*] |
| PPOR | MMM+6 | MMM+4 | MMM+7.5 | 34.5 | 32.5 | 36 | 34.3 | NA |

Notes.

ACER, *Acropora cervicornis*; DLAB, *Diploria labyrinthiformis*; MCAV, *Montastraea cavernosa*; OANN, *Orbicella annularis*; OFAV, *Orbicella faveolata*; PAST, *Porites astreoides*; PPOR, *Porites porites*.

MMM refers to the maximum monthly mean temperature of 28.5 °C. $F_v/F_m$, photosynthetic efficiency.

[*]Modified from the actual $T_{50}$ average across methods.

NA corresponds to species for which a second phase was not conducted.

all temperatures (Appendix S1: Table S5). Additional species-specific differences were observed at individual temperature levels, most frequently involving *A. cervicornis* and one or more other species (Appendix S1: Table S3). The temperature that elicited greater differences between species corresponded to 34.5 °C (MMM+6). Median values of pixel intensity (Appendix S1: Table S6) revealed similar thresholds across species around 34 °C (Table 1). The highest threshold value (34.5 °C) was shared by *A. cervicornis*, *M. cavernosa*, and both *Orbicella* spp., whereas the lowest ones corresponded to *Porites* spp.

### Bleaching visual rank

For most species total bleaching was evident at 36.5 °C (MMM+8), except for *D. labyrinthiformis* which started at 34.5 °C, and *P. porites* which did not show total bleaching until 37.5 °C (MMM+9) (Fig. 2C). Similarly, temperatures that elicited the most variable bleaching responses across species were 35.5 °C and 36 °C (MMM+7 and +7.5, respectively) except for *D. labyrinthiformis* (34 °C, MMM+5.5) (Table 1).

### Final $T_{50}$ determination across methods

$T_{50}$ temperatures varied between species and generally ranged between 34 and 36 °C (Table 1). The $T_{50}$ values derived from different methods also showed varying degrees of consistency. For *D. labyrinthiformis*, thresholds were identical across methods (34 °C), while for *A. cervicornis* there was a 1 °C difference between one of the proxies (visual bleaching at 35.5 °C) and the other two (both 34.5 °C). Thresholds for *M. cavernosa* and both *Orbicella* species ranged within 0 to 1.5 °C across methods. In contrast, *P. astreoides* and *P. porites* showed greater variation, with method-derived thresholds ranging across 1.5 °C and up to 3.5 °C, respectively.

The $T_{50}$ selected for the intraspecific phase of *Orbicella* spp. and *P. astreoides* were adjusted from the arithmetic mean of $T_{50}$ values across methods based on the following rationale: (1) To ensure that variability in visual bleaching rank was captured; (2) At

35.5 °C, broad variation in photosynthetic efficiency responses was still observed for the three species; and (3) pixel intensity values at this temperature remained within a moderate range relative to the overall distribution, with average values of 100.32 ± 16.43 for *O. annularis* (median = 68), 101.39 ± 16.42 for *O. faveolata* (median = 86.41), and 112.41 ± 17.99 for *P. astreoides* (median = 88.36) (Appendix S1: Table S6). These values suggest bleaching was in progress while still allowing for variation across genotypes to be observed.

Principal component analysis demonstrated that photosynthetic efficiency and bleaching visual rank were highly correlated (Spearman's rank; $\rho = 0.8$, $p = 1.729e{-}13$) whereas pixel intensity and bleaching visual rank were moderately correlated (Spearman's rank; $\rho = 0.6$, $p = 8.537e{-}07$). No correlation was found between photosynthetic efficiency and pixel intensity (Spearman's rank; $\rho = -0.3$, $p = 0.0229$). The first principal component (PC1) explained 69% of the variance, with bleaching visual rank and photosynthetic efficiency accounting for most of the performance success attributed to the PC1 (factor loadings of 0.67 and 0.59, respectively). The performance of the subjects (one species under a specific temperature treatment) was visually organized into three groups along the $x$-axis or PC1 (Fig. 3). The best performers on the left of the $x$-axis mostly corresponded to all species at lower temperatures. In contrast, the worst performers to the right of the $x$-axis corresponded to some species at the highest temperatures. The middle performers' group showed some species and temperatures corresponding to the $T_{50}$ thresholds selected for the *Orbicella* spp. and *M. cavernosa* (Table 1). The second principal component (PC2) corroborated *A. cervicornis* performance differences with the rest of the species across temperatures and methods, which is reflected as a division into two groups along the $y$-axis (Fig. 3). A PERMANOVA test confirmed that the performance categories identified visually in the PCA corresponded to statistically distinct multivariate stress responses ($F = 13.49$, $R^2 = 0.34$, $p = 0.001$).

## Intraspecific response (Phase 2)
### Thermotolerance of colonies across methods

*Orbicella* spp. colonies showed a wide distribution of photosynthetic efficiency values: three colonies exhibited the highest thermotolerance ($F_v/F_m > 0.5$), four showed high thermotolerance (0.4–0.5), two were in the lowest category ($F_v/F_m < 0.3$), and the remaining colonies had values between 0.3 and 0.4 (Fig. 4A). *Montastraea cavernosa*, in contrast, showed a narrower distribution: four colonies had $F_v/F_m > 0.5$, two colonies were in the 0.3–0.4 range, and the majority of the colonies fell between 0.4 and 0.5. The temperatures selected for *D. labyrinthiformis* and *P. astreoides* did not lead to notable impairment of the photosynthetic system across colonies, as all individuals exhibited $F_v/F_m$ values above 0.4. Thermotolerance scores for these species were only distributed across the two highest categories, with colonies showing either high (0.4–0.5) or the highest (>0.5) photosynthetic efficiency (Fig. 4A). Pixel intensity ratios of heated samples relative to their respective controls revealed both the top and bottom 25% of colonies in terms of bleaching severity for all species (Fig. 4B). *Porites astreoides* exhibited the lowest variability in bleaching visual scores, with all but one colony falling within a single bleaching category

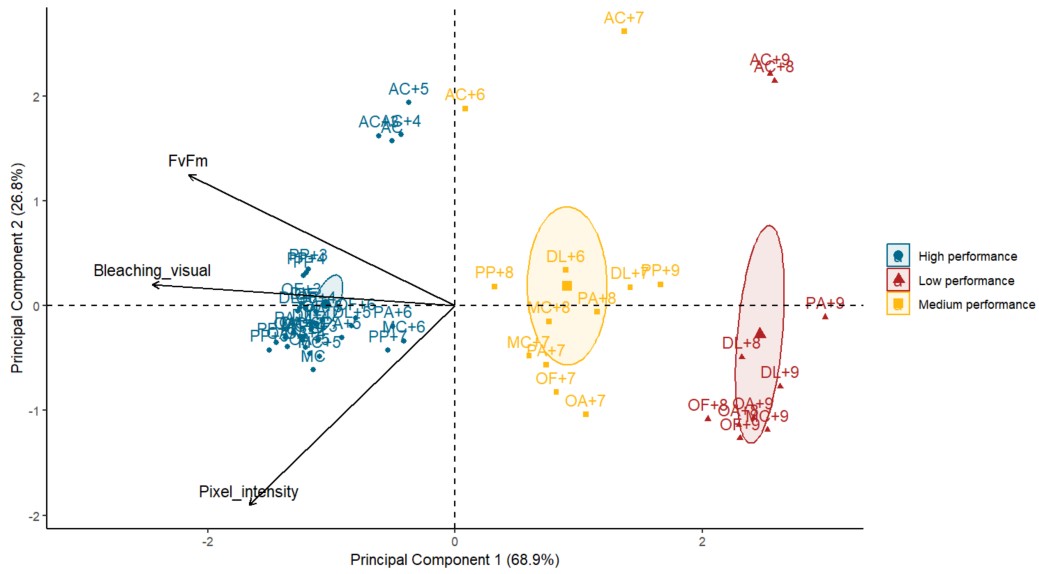

**Figure 3  Biplot of principal component analysis (PCA) summarizing heat stress performance across three evaluation methods for each coral species under varying temperatures.** Each point represents the averaged performance of a species at a given temperature across all methods. Points are grouped by performance categories with confidence ellipses illustrating the clustering of similar responses. Arrows indicate the contribution and direction of each variable to the PCA axes, with principal components 1 and 2 explaining 68.9% and 26.8% of the variance, respectively. AC, *Acropora cervicornis*; DL, *Diploria labyrinthiformis*; MC, *Montastraea cavernosa*; OA, *Orbicella annularis*; OF, *Orbicella faveolata*; PA, *Porites astreoides*; PP, *Porites porites*. Temperatures are coded with number of degrees celsius above the maximum monthly mean temperature (MMM) of 28.5 °C. Species name with no "+" temperature code indicate the MMM temperature.

(low; scores between 0.3–0.4) (Fig. 4C). In contrast, both *Orbicella* spp. showed high variability, with colonies more evenly distributed across all four bleaching categories, indicating a broader range of visual bleaching responses.

## Thermotolerance final rank

Scores across methods were added to obtain a single thermotolerance total score per colony (Fig. 5). Colonies #03 and #07 of *D. labyrinthiformis* were excluded from the visual bleaching score as only two heated fragments were available, due to limited tissue on the colony. While photophysiology and pixel intensity scores were averaged from the four available fragments, the visual metric required four heated replicates and was thus not applied. Figure 5 showed overall variability in the bleaching response within each species, and revealed the best thermotolerance performers that could be prioritized for restoration. The principal component analysis showed that visual bleaching, followed closely by pixel intensity, explained most of the variance (factor loadings of 0.7 and 0.69, respectively), while photosynthetic efficiency explained the least with a coefficient of 0.14 (Appendix S1: Figs. S2 and S3). The scores obtained through the PC1 and those obtained through the scoring procedure for each method and its subsequent addition used in this study showed a positive correlation (Spearman's rank; $\rho = 0.84$, $p < 2.2e{-}16$) (Fig. 6). Thermotolerance

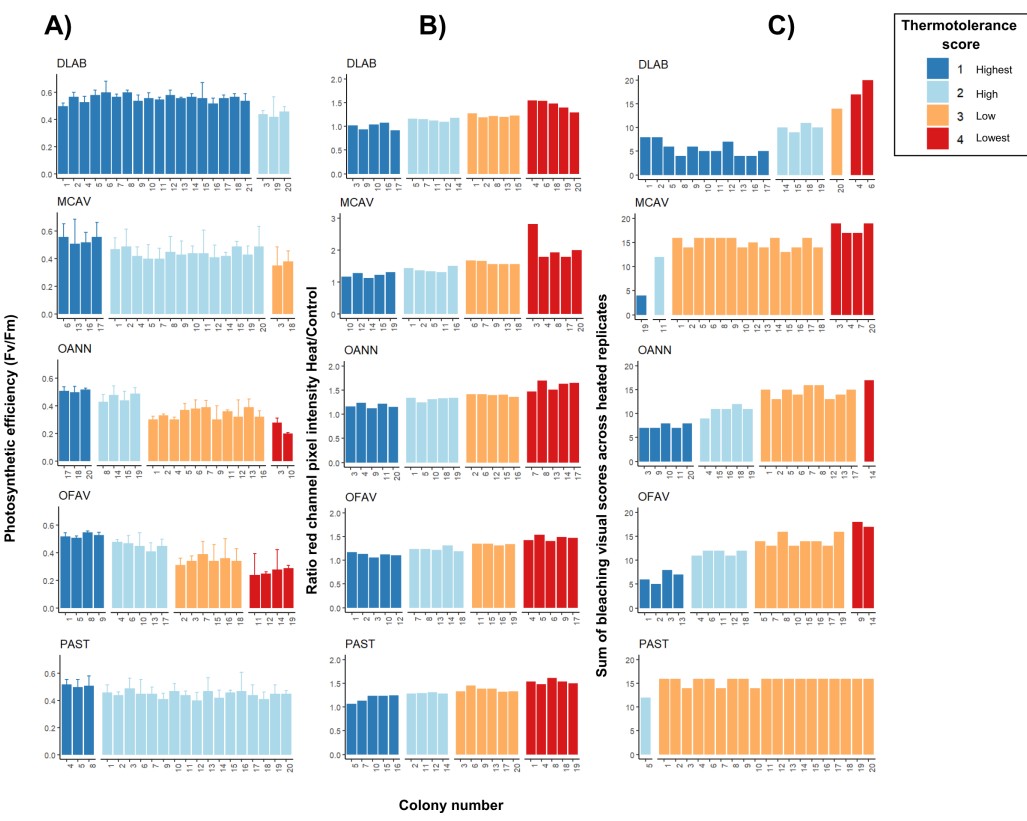

**Figure 4 Intraspecific heat stress response results of photosynthetic efficiency ($F_v/F_m$) (A), pixel intensity analysis (B), and bleaching visual rank (C).** Four replicates per treatment (control and heat under $T_{50}$) were used per each colony. Results of both $F_v/F_m$ (A) and bleaching visual scores (B) show only heated replicates ($n = 4$); whereas pixel intensities were calculated as a ratio of heated over control replicated ($n = 4$ for each ratio and colony). DLAB, *Diploria labyrinthiformis*; MCAV, *Montastraea cavernosa*; OANN, *Orbicella annularis*; OFAV, *Orbicella faveolata*; PAST, *Porites astreoides*. Note different values in $y$-axis of B) for MCAV due to higher ratios.

scores raised the question of whether coral performance might be influenced by the reef of origin. Although initial visual inspection suggested a potentially lower performance of *M. cavernosa* colonies from Magayan reef, thermotolerance scores varied across all reefs and species without consistent patterns for any particular site or taxon (Appendix S1: Fig. S4). This observation was supported by statistical analysis, which found no significant differences in thermotolerance scores among reefs of origin for any species (Kruskal–Wallis; $p > 0.05$; Appendix S1: Table S7).

## DISCUSSION

This study introduces a novel, practical framework that integrates species-specific thermal thresholds with intraspecific performance data to identify thermotolerant coral colonies for restoration. Our approach emphasizes context-specific thresholds tailored to local environmental and biological conditions. By combining multiple proxies under ecologically

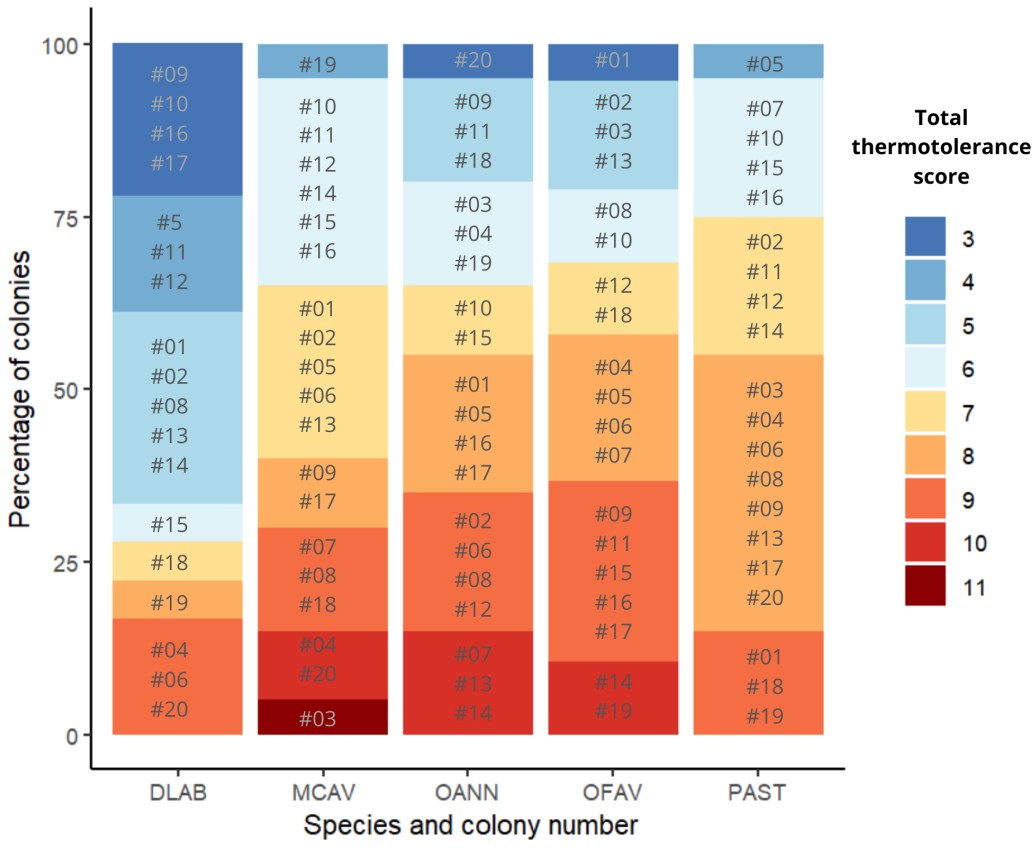

**Figure 5  Ranking of colonies of the five species ordered from best to worst performance across methods (lowest total thermotolerance scores in blue, and highest total thermotolerance scores in red, respectively).** DLAB, *Diploria labyrinthiformis*; MCAV, *Montastraea cavernosa*; OANN, *Orbicella annularis*; OFAV, *Orbicella faveolata*; PAST, *Porites astreoides*.

relevant heat stress, the framework addresses key gaps in coral thermotolerance assessment and supports more targeted, effective restoration planning.

## Interspecific response (Phase 1)
### Comparative and contextual thermal resilience across species

Differences in the heat stress response were observed between species. *Montastraea cavernosa* and *A. cervicornis* exhibited the highest $T_{50}$ value across methods, followed by the *Orbicella* and *Porites* spp. On the other hand, *D. labyrinthiformis* showed the lowest $T_{50}$ (averaged $T_{50}$, Table 1). A similar response across methods was observed between *Orbicella* congenerics. These results are in agreement with other studies and field observations where *M. cavernosa* showed less susceptibility to heat stress in comparison to *Orbicella* spp. (*Fitt & Warner, 1995*; *Swain et al., 2017*), followed by *D. labyrinthiformis* and branching *Porites* spp. (*Smith et al., 2013*; *Alemu & Clement, 2014*).

However, this pattern contrasts with previous results for *P. astreoides*, which is generally considered one of the least affected species by laboratory-induced thermal stress and bleaching events in the region (*Warner et al., 2006*; *Lima, Bursch & Dinsdale, 2022*). *Porites*

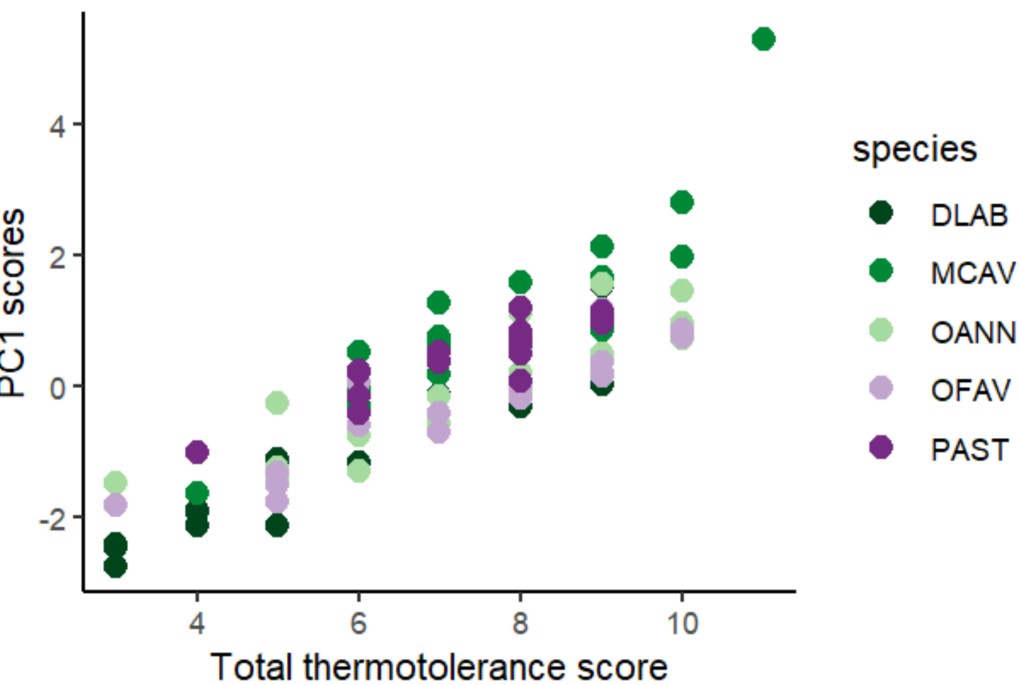

**Figure 6** **Positive correlation between PC1 loadings of the principal component analysis of all colonies across species and methods, and final thermotolerance scores across methods.** DLAB, *Diploria labyrinthiformis*; MCAV, *Montastraea cavernosa*; OANN, *Orbicella annularis*; OFAV, *Orbicella faveolata*; PAST, *Porites astreoides*.

*astreoides* represents one of the species with the highest relative cover on Caribbean reefs (*Green, Edmunds & Carpenter, 2008*), and its abundance has been enhanced by its relatively low susceptibility to coral disease outbreaks, along with congeneric species (*Aronson & Precht, 2001*; *Precht et al., 2016*). However, it has also been affected during recent bleaching events (*Manzello, Berkelmans & Hendee, 2007*; *Smith et al., 2019*; *Edmunds, Didden & Frank, 2021*), and its recovery capacity has been limited compared to that of species such as *O. faveolata* (*Grottoli et al., 2014*). This limited resilience has been linked to an irreversible decline in carbohydrates and proteins following heat stress, as well as a reduced ability to shift its endosymbionts (*Grottoli et al., 2014*; *Schoepf et al., 2015*). The results of this study support predictions that despite their widespread presence on Caribbean reefs, the success of *P. astreoides* may be compromised in the long term assuming that bleaching events continue to intensify (*Levas et al., 2018*; *Lima, Bursch & Dinsdale, 2022*).

In the present study, *P. porites* exhibited one of the lowest average $T_{50}$ values (Table 1). However, this value was largely influenced by a low $T_{50}$ obtained through pixel intensity analysis, which could have underestimated bleaching severity since pixel values increased again at higher temperatures. Moreover, the visual bleaching threshold for this species was one of the highest observed, with no sign of symbiont physiological collapse even at elevated temperatures—levels at which such collapse was evident in more thermotolerant species of the study like *M. cavernosa*.

Many studies that highlight the greater vulnerability of branching *Acropora* species compared to massive and encrusting taxa have been conducted in the Red Sea and Indo-Pacific (*Loya et al., 2001*; *Pratchett et al., 2013*; *Harrison et al., 2019*). However, similar patterns have also been reported in the Caribbean (*e.g.*, *Langdon et al., 2018*; *Cramer et al., 2021*; *Palacio-Castro et al., 2021*). In this study, *A. cervicornis* was among the most thermotolerant species, showing no collapse in the photosynthetic efficiency of its symbionts. This could be the result of strong natural selection among Caribbean genotypes following repeated disease and bleaching events (*Aronson & Precht, 2001*; *Cramer et al., 2020*). This is consistent with *Manzello, Berkelmans & Hendee (2007)*, who reported low susceptibility of *A. cervicornis* in certain sites.

Recent studies in the Indo-Pacific have also suggested shifts in susceptibility patterns. *Guest et al. (2016)* observed higher tolerance in *Acropora* and *Pocillopora* during a bleaching event, whereas *Porites* species were more affected. Similar findings were reported by *Szereday, Voolstra & Amri (2024)*. *Burn et al. (2023)* further observed that massive *Porites* exhibited high bleaching susceptibility in sites with greater overall bleaching severity across the Great Barrier Reef and Coral Sea, while *Acropora* species were also highly affected under these conditions. However, *Porites* and small *Acropora* colonies were identified among the "short-term winners" in *Van Woesik et al. (2011)*, suggesting that bleaching outcomes can vary depending on colony size, morphology and environmental context. Conversely, some Caribbean studies have documented lower heat stress tolerance in *A. cervicornis* compared to *O. faveolata* (*Langdon et al., 2018*). Notably, this study involved a long-term experimental design, highlighting that species may respond differently depending on the type and duration of stress.

Therefore, conclusions about relative susceptibility or resilience should be drawn cautiously, considering the evaluation method and specific environmental context, as responses may also vary between reef sites (*Manzello, Berkelmans & Hendee, 2007*; *McClanahan et al., 2020*). Each species may exhibit distinct tolerance and recovery mechanisms (*Swain et al., 2016*; *Thomas et al., 2019*; *Conti-Jerpe et al., 2020*), but this can also differ among individuals and be shaped by environmental conditions (*Thomas et al., 2018*; *Drury & Lirman, 2021*; *Ruggeri et al., 2024*).

Interactions between host genetic and physiological factors, microbial and algal associated communities (*Santoro et al., 2025*), and other abiotic and biotic factors of a determined environment underscore the complexity of thermotolerance. No data on symbionts or the microbiome was collected for this study, so the potential influence of these factors on the thermotolerance of local individuals remains unknown. However, a recent study by *O'Donnell et al. (2024)*, which analyzed coral populations from the same reefs of the present study, found that the symbiont community composition varied significantly among species of the genus *Orbicella*, while the symbiont communities of *P. astreoides* from different reefs were mostly dominated by *Symbiodinium* with little variation. These findings are consistent with previous literature showing *Orbicella* spp. hosting diverse *Symbiodinium* clade B genotypes, often including multiple B1 haplotypes with geographic structuring but relatively consistent distribution across species (*Green et al., 2014*). Additional studies in Florida have shown that *Orbicella* spp. can host both *Breviolum* and *Cladocopium*,

occasionally shifting their symbiont communities under environmental stress (*Dennison et al., 2021*). In contrast, *P. astreoides* populations tend to maintain associations with a single dominant *Symbiodinium* lineage across sites (*Kenkel et al., 2013*). In this context, the importance of establishing thresholds at a local level and identifying specific thermotolerant colonies for each target species is highlighted (*Van Woesik et al., 2011*), especially given the strong interplay between host identity, symbiont composition, and thermal sensitivity.

### Temperature thresholds

Although species differed in their heat stress responses, threshold temperatures generally fell within a similar range of approximately 34 to 36 °C (MMM + 5.5 to +7.5) across all corals tested under this experimental heat pulse design. This clustering of $T_{50}$ values is notable, especially considering that $ED_{50}$ values reported in other studies can vary widely within the same species. These differences are often due to factors like local environmental history, genetic variation, or differences in how the data are analyzed. In contrast, the relatively consistent $T_{50}$ range found here may reflect the ability of the standardized heat ramp protocol to identify moderate and ecologically meaningful stress points under controlled conditions. These values align with prior studies using comparable temperature profiles, such as *Cunning et al. (2021)*, who demonstrated an $ED_{50}$ range of 35 to 37 °C for *Acropora cervicornis* based solely on photosynthetic efficiency, and *Klepac et al. (2024)*, who found an $ED_{50}$ of 34.3 °C for the same species. Notably, our study identified a similar $T_{50}$ of 34.5 °C for *A. cervicornis*, reinforcing the relevance of the approach. All three studies applied heat stress protocols consistent with the CBASS framework, which has become increasingly used globally due to its standardized experimental design (*Voolstra et al., 2020*; *Evensen et al., 2021*; *Evensen et al., 2022*; *Evensen et al., 2023*).

In some cases, such as for *A. cervicornis*, *Orbicella annularis*, and *Porites porites*, the $T_{50}$ value preceded a marked decline in bleaching metrics (Fig. 2). This pattern likely resulted from the discrete set of temperatures tested and reflects the applied nature of $T_{50}$: it was selected to identify a moderate and variable response across genotypes, rather than to determine a precise physiological tipping point. While $ED_{50}$-based approaches aim to define species-wide tipping points through curve fitting, the $T_{50}$ approach serves a complementary purpose by emphasizing temperatures that capture intraspecific variation, thereby informing genotype-level selection for restoration.

Importantly, the thresholds identified in this study should not be interpreted as universal for each species. Rather, they represent context-dependent responses shaped by local environmental and biological factors (*Fitt et al., 2001*; *Suggett & Smith, 2020*), and may also shift over time due to acclimatization, phenotypic plasticity, or environmental change. Nonetheless, $T_{50}$ values remain a valuable comparative metric when interpreted within the boundaries of the experimental design. By helping to characterize species- and genotype-specific responses under controlled but ecologically relevant conditions, these thresholds can support improved predictions and decision-making in restoration planning across different geographic and environmental contexts.

## Intraspecific response (Phase 2)

Variability in the stress response was observed across genotypes, to a greater or lesser extent, depending on the species-selected $T_{50}$ combination and the stress evaluation method. High variability in photosynthetic efficiency across genotypes was only observed for *Orbicella* spp. For *D. labyrinthiformis*, *M. cavernosa,* and *P. astreoides,* the temperature selected as $T_{50}$ seemed insufficient to elicit variability in photosynthetic efficiency within the current experimental design. In the case of *M. cavernosa*, this could be because the $T_{50}$ calculated for the photosynthetic efficiency in the interspecific phase was 0.5 °C higher than that selected across methods for the intraspecific phase. However, this is incongruent with the cases of *D. labyrinthiformis* and *P. astreoides* since the final $T_{50}$ selected in these species was higher than that established for their photosynthetic efficiency. A similar case was observed for *D. labyrinthiformis* during the bleaching visual ranking, where low values of bleaching were recorded across colonies in the intraspecific phase despite using the same $T_{50}$ determined for bleaching visual in the interspecific phase. This may be since for both phases the selected colonies were different (in the first phase they were selected at random and in the second phase, specific parent colonies were used), so for future experiments, it is recommended to use the same population sample for both phases, if possible, without compromising colony health or the availability of tissue needed for propagation purposes.

Additionally, this study did not consider seasonality (Appendix S1: Table S1), which is known to influence physiological traits (*Fitt et al., 2001*; *Chapron et al., 2022*). This omission may have affected the temperature differences needed to trigger response variability in the species *D. labyrinthiformis* and *P. astreoides*, with assessments conducted in April and October for Phase 1 and in November and March for Phase 2, respectively. In contrast, both experimental phases for *M. cavernosa* were conducted in the same season (May and June). Meanwhile, *Orbicella* spp. exhibited intraspecific variability across methods at the established $T_{50}$, even though their experimental phases took place in different seasons (April and May for Phase 1, and November and February for Phase 2 for *O. faveolata* and *O. annularis*, respectively). This underscores that, while seasonality is a crucial factor, many other elements may significantly influence stress response variability, suggesting that they should be prioritized when establishing a temperature threshold. Since each species was assessed only once during each experimental period, the effects of season and species are confounded and cannot be statistically disentangled. Therefore, while standardized experimental conditions were used, we acknowledge that this does not eliminate potential underlying seasonal influences.

*Montastraea cavernosa* and *P. astreoides* showed severe levels of visual bleaching, but no physiological stress was observed across the majority of genotypes. This highlights the need of identifying a temperature 'sweet spot' that demonstrates variability based on the evaluation method (see next section). Notably, both *Orbicella* spp. exhibited variability across methods and genotypes, successfully revealing an optimal $T_{50}$ specifically tailored for their population. It should be noted that finding variability is also influenced by the scoring criteria of each method, since for instance, in the case of pixel intensity analysis, the division of the data into quartiles will always guarantee that there is representativeness of each thermal score for each set of genotypes. Obtaining variability will also depend on the

selected genotypes, and may be biased if there is not a lot of variability in the sample. This could be due to a low population genetic diversity, the adaptation of several genotypes to a similar microenvironment, or the hosting of a similar type or community of symbionts (*Sturm et al., 2020*; *Aguilar et al., 2024*; *O'Donnell et al., 2024*). Therefore, it is necessary to find the optimal number of individuals that ensure a greater variability of responses, taking into account the environmental, logistical, and governmental limitations of each locality. For instance, *Shearer, Porto & Zubillaga (2009)* suggested that 10 donor colonies are sufficient to retain over 50% of the genetic diversity of a population, while 35 donor colonies are needed to preserve more than 90% of the genetic diversity in an average coral population. Also, *Baums et al. (2019)* proposed that 20–25 genets per species including three to six genets per reef along environmental gradients would capture most intraspecies genetic diversity and would represent diverse environmental conditions in the area.

## Heat stress evaluation methods

Throughout the results of this study, differences were observed in both inter- and intraspecific responses depending on the stress evaluation method. This is not surprising, as each proxy targets a different physiological or visual aspect of the coral bleaching process. While $F_v/F_m$ measures physiological stress from photoinhibition of symbionts at a given point in time, visual bleaching (assessed by visual ranking and pixel intensity) reflects cumulative tissue color loss, which may be caused by reductions in symbiont density, chlorophyll content, or both (*Fitt et al., 2001*). In addition, thermal stress can precede visible bleaching (*Lesser, 2021*), making the timing of measurement a key consideration when interpreting results (*Nielsen et al., 2022*). This justifies the use of multiple, complementary methods when evaluating coral thermotolerance.

Combining these proxies enabled a more holistic evaluation of bleaching responses. For instance, in the interspecific heat response of *P. porites* a lack of photosynthetic efficiency crash was found at the highest temperatures while pixel intensities demonstrated bleaching at lower temperatures compared to the other species. In addition, the fact that no physiological stress was observed in the intraspecific response of *D. labyrinthiformis* and *M. cavernosa* while presenting high visual bleaching susceptibility implies that perhaps the selected $T_{50}$ temperature should have been higher at the expense of measuring color bleaching earlier. These discrepancies indicate that relying on a single method could misrepresent the true extent of stress. *Klepac et al. (2024)* also observed genotype-level discrepancies between physiological and visual traits (chlorophyll and protein content), highlighting the complexity of the heat stress response (*Weis, 2008*), supporting the need for multi-metric assessments (*McClanahan et al., 2020*).

Although the three methods differ in their level of subjectivity and standardization, equal weighting of all three proxies in determining $T_{50}$ and thermotolerance rankings was chosen to reflect the complementary nature of the information they provide. Rapid photodamage is captured by $F_v/F_m$, progressive pigment loss by pixel intensity; and fast, field-relevant assessments by visual scores. Given the practical goal of supporting restoration decisions, prioritizing methodological complementarity over hierarchical weighting provides a more robust and applicable framework.

Visual bleaching scores, while more subjective, proved informative for rapid assessment of colony performance, as would be needed in restoration settings. However, their interpretation may be influenced by morphological traits such as tissue thickness and skeletal porosity, which were not quantified here but are known to affect bleaching visibility (*Todd, 2008*). For example, thinner-tissued or denser-skeleton species may display more apparent color change than thicker-tissued or more porous ones. This underscores the need to interpret visual data cautiously, especially under short-term, high-temperature exposures, and highlights the importance of integrating morphological information into future assessments. In addition, pixel intensity results can be biased by the presence of endolithic algae, which darken the skeleton and reduce apparent bleaching (*Pernice et al., 2020*). However, this distinction can often be made by visual inspection. A clear example of this can be seen in both Fig. 2B and Fig. S2 (Appendix S2), where *A. cervicornis* presented the brightest intensities as its skeleton appeared to host fewer endolithic algae compared to the other species (*Ricci et al., 2019*).

Altogether, combining non-invasive proxies such as photosynthetic efficiency and pixel intensity allows for a more refined understanding of coral health under heat stress. These methods have already been identified as ideal rapid tests of thermotolerance (*Nielsen et al., 2022*), and integrating them with visual scores strengthens result interpretation. While traits such as symbiont density and chlorophyll content often correlate with photophysiology, incorporating additional markers like antioxidant enzyme activity, host protein content, or heat shock proteins may further enhance assessments of bleaching severity and recovery potential (*Scheufen, Iglesias-Prieto & Enríquez, 2017*; *Voolstra et al., 2020*; *Colín-García et al., 2024*).

## Selection of thermotolerant colonies

The present study proposes a framework for selecting coral colonies based on their natural thermotolerance to assist restoration practitioners and conservationists in making strategic decisions that enhance ecosystem resilience in light of the increasing frequency of marine heat waves caused by climate change. In turn, this study could contribute to the standardization of selection practices for potentially more thermotolerant colonies and provide relevant information for a better understanding of coral thermal stress (*Caruso, Hughes & Drury, 2021*). The final product of this framework is a ranking of colonies of different species ordered from best to worst performance based on their photosynthetic efficiency and visual bleaching (Fig. 5) measured in a heat pulse stress assay with a standardized temperature profile (*Voolstra et al., 2020*; *Evensen et al., 2021*; *Evensen et al., 2022*; *Evensen et al., 2023*).

The scoring procedure based on the different heat stress traits is validated by the positive correlation found between the scores and the loadings of the PC1 of the PCA of all colonies across species and methods (Fig. 6). Based on this colony ranking, restoration practitioners and managers can decide which colonies are to be selected for reproduction and/or transplantation (*i.e.,* the 10% or the 25% top performers). It is important to note that thermotolerance, as assessed here, is inherently a relative trait, context-dependent by nature, since the classification of colonies into performers and non-performers is always
made in reference to the sample set tested. As such, rankings may not always be directly comparable across experiments or populations. However, they remain highly informative for identifying the most robust candidates within a given environmental and experimental context, which aligns with the applied goals of restoration. In cases where the variability of responses is low, it would be advisable to repeat the procedure with a larger sample size, or from more diverse environments (different depths, thermal histories, current exposure, *etc.*) (*Bayraktarov et al., 2013*), always taking into account the environmental conditions of the destination of these colonies.

In the Caribbean, some studies have already conducted thermotolerant colony selection for the species *A. cervicornis* (*Cunning et al., 2021*; *Klepac et al., 2024*). However, the present study constitutes a novel advance as it represents the first selection of thermotolerant colonies for non-branching coral species, including columnar, mounding, boulder, and submassive forms, in the Caribbean.

### Future research

In addition to fine-tuning colony selection under laboratory conditions by standardizing a procedure that takes into account the population and environmental variability of different localities (*Voolstra et al., 2020*), experiments should be supported with more evidence on the in-wild performance of previously selected colonies (*Morikawa & Palumbi, 2019*; *Klepac & Barshis, 2022*) and on how the environment can influence thermotolerance traits (*Howells et al., 2013*; *Towle et al., 2017*; *Drury & Lirman, 2021*). For instance, *Alderdice et al. (2024)* proved that the thermotolerance of *Acropora hyacinthus* (Dana, 1846) colonies was reduced one year after transplantation. In this sense, further study of phenotypic plasticity including a temporal component is needed, with studies such as those carried out by *Kenkel & Matz (2016)* with *Porites astreoides*, and more recently by *Castillo et al. (2024)* with *Siderastrea siderea* (Ellis & Solander, 1786); with the aim of better understanding the relationship between phenotypic plasticity and thermotolerance.

Furthermore, phenotypic plasticity and recovery capacity after bleaching play an important role in structuring future coral communities (*Grottoli et al., 2014*; *Matsuda et al., 2020*), especially considering that today's bleaching thresholds may not be the same as in the future. Therefore, there is also a need to better understand how bleaching thresholds extrapolate to nature and which metrics are best to predict local bleaching events (*Klepac & Barshis, 2022*) and make better-informed conservation and restoration decisions.

Ultimately, a critical research question remains: will corals identified as thermally tolerant under experimental conditions actually maintain their performance once outplanted, and do they outperform non-selected corals in real reef environments over time?

## CONCLUSIONS

The present study addresses key knowledge gaps regarding coral thermotolerance by establishing context-specific temperature thresholds and developing a standardized framework to support the selection of resilient colonies for restoration. Findings revealed distinct thermal response patterns across species and among genotypes within species,

demonstrating that thermotolerance is shaped by both interspecific and intraspecific variation. Notably, *Acropora cervicornis* and *Montastraea cavernosa* exhibited the highest thermotolerance under the experimental conditions.

By integrating practical short-term assays with multiple evaluation methods, this study offers a replicable tool to inform colony selection in restoration efforts. However, the framework is not intended to define fixed, universal thresholds, but rather to support localized, comparative assessments that reflect the ecological, biological, and geographic context of each population. Interpreting species hierarchies or individual performance scores requires consideration of factors such as environmental history, phenotypic plasticity, and the time frame of measurement.

The study underscores the value of a multi-metric approach, as it provides a more holistic understanding of coral health and thermotolerance. While $T_{50}$ values serve as useful reference points, they are operational tools rather than absolute predictors and may shift over time or across environments.

Limitations such as potential seasonal influences and the absence of microbial and symbiont data underline areas for improvement in future experiments to better understand thermotolerant mechanisms in specific contexts. Additionally, integrating physiological and molecular markers could refine thermotolerance assessments and enhance predictive models. Finally, long-term monitoring of selected colonies in natural settings could support the evaluation of persistent thermotolerance traits and the role of plasticity over time. Altogether, this study proposes a flexible yet standardized selection framework that supports reef restoration through the identification of thermotolerant genotypes, offering a promising pathway for enhancing reef resilience in the face of accelerating climate change.

## ACKNOWLEDGEMENTS

We thank Dr Steve Palumbi for leading the first thermotolerance experiment in the Dominican Republic at the Iberostar Coral Lab in 2019, laying the groundwork for this study. We also thank the Dressel Divers team for their assistance in diving operations. Finally, we thank the Dominican Republic's Ministry of Environment for providing the necessary permits for coral sampling and research.

### Funding

Research work was supported by Iberostar Group through Wave of Change Innovation Hub. Publication fees were covered by the University of Groningen. The funders had no role in study design, data collection and analysis, decision to publish, or preparation of the manuscript.

### Grant Disclosures

The following grant information was disclosed by the authors:
Iberostar Group.
University of Groningen.

## Competing Interests

The authors declare there are no competing interests. Macarena Blanco Pimentel and Johanna Calle Triviño are employed by Iberostar Group. Megan K. Morikawa is employed by Salesforce.

## Author Contributions

- Macarena Blanco Pimentel conceived and designed the experiments, performed the experiments, analyzed the data, prepared figures and/or tables, authored or reviewed drafts of the article, and approved the final draft.
- Johanna Calle-Triviño performed the experiments, authored or reviewed drafts of the article, and approved the final draft.
- Daniel J. Barshis conceived and designed the experiments, authored or reviewed drafts of the article, and approved the final draft.
- Sancia E.T. van der Meij analyzed the data, prepared figures and/or tables, authored or reviewed drafts of the article, and approved the final draft.
- Megan K. Morikawa conceived and designed the experiments, authored or reviewed drafts of the article, and approved the final draft.

## Field Study Permissions

The following information was supplied relating to field study approvals (i.e., approving body and any reference numbers):

Coral sample collection and experiments were carried out under all the required research permits, as well as under the contract for access to genetic resources between Iberostar and the Ministry of the Environment of the Dominican Republic, which complies with the legal framework of the Protocol of Nagoya and the Convention on Biological Diversity.

## Data Availability

The data is available at figshare: Blanco Pimentel, Macarena; Calle-Triviño, Johanna; Barshis, Daniel; E.T. Van Der Meij, Sancia; Morikawa, Megan K. (2025). Building heat-resilient Caribbean reefs: Integrating thermal thresholds and coral colonies selection in restoration. figshare. Dataset. https://doi.org/10.6084/m9.figshare.28292339.v1.

## Supplemental Information

Supplemental information for this article can be found online at http://dx.doi.org/10.7717/peerj.19987#supplemental-information.

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
