# Peer review of "Building heat-resilient Caribbean reefs: integrating thermal thresholds and coral colonies selection in restoration"

_PeerJ, doi:10.7717/peerj.19987_

## Round 0.1 · original submission · Major Revisions

This is an important manuscript examining coral thermal tolerance in the Dominican Republic. There is a general lack of research done on these reefs, and a lack of representation of Dominican scientists in reef research, both of which are important reasons that I believe PeerJ should promote this work. I also request that the authors make this point clear in the title/abstract and introduction to bring attention to Dominican reefs as part of the Caribbean complex deserving of research and restoration attention. The work was reviewed by two experts in the fields of reef restoration and thermal tolerance, both of whom provided measured, detailed and fair reviews. The central theme of the work, addressing the development of thermal tolerance thresholds, is a priority for restoration, and I agree firmly with Reviewer 2 that the authors are responsible to place their work in the context of the trends toward a standardized approach to thermal tolerance estimations using CBASS systems. I acknowledge, as does the reviewer, that no approach is perfect and alternatives are valid, but it is important to contextualize alternative approaches to ensure interpretability and understanding. I look forward to a thoroughly revised manuscript addressing each of the issues raised by the reviewers as well as those I have outlined here, and consider these Major Revisions.

·

Basic reporting

I found this study to be quite strong. This manuscript has a clever study design, is well written and well structured, and author assertions are supported with abundant references. My main overarching feedback would be that the discussion could be restructured a bit for clarity. There are a lot of well supported assertions (and qualifying statements), but they could be ordered into a bit more of a logical flow (within the sections of the discussion you already have) to more clearly drive home your key findings.

See attached doc for specific edits/suggestions

Experimental design

This study investigates interspecific variation in bleaching for multiple species of Caribbean corals using three indices: Fv/Fm, pixel intensity, and visual scores of bleaching, and identifies critical thermal thresholds for each species. The authors then use this threshold to quantify intraspecific variation of thermal tolerance, identifying thermally-tolerant individual corals that would be well suited for bleaching.

Validity of the findings

See attached doc for specific edits/suggestions

Additional comments

See attached doc for specific edits/suggestions

Reviewer 2 ·

Basic reporting

No comment

Experimental design

no comment

Validity of the findings

While the study addresses an important goal—assessing coral heat tolerance across species—I have concerns about the analytical approach, particularly the substitution of standard ED50 metrics with medians and the rationale for combining three different assessment methods. These choices reduce the robustness and comparability of the results. The experimental design aligns closely with the established CBASS framework, but the manuscript does not clearly justify modifications to this method. With revisions to the data analysis, statistical modeling, and a clearer rationale for methodological choices, the study could offer meaningful replication and insight.

Additional comments

In their paper "Building heat-resilient Caribbean reefs: Integrating thermal thresholds and coral colonies selection in restoration," the authors use the CBASS assay design and combine Fv/Fm measurements, pixel intensity, and visually assigned bleaching scores to assign heat-tolerance thresholds for seven coral species. These thresholds are then used to assess thermotolerance in genets (pre-tagged colonies). This is an important and timely study: as bleaching events increase in frequency and severity, the role of coral nurseries in reef restoration becomes more critical. Using rapid, standardized, and effective methods to identify heat-tolerant individuals is urgently needed. The authors’ inclusion of multiple species and use of a widely recognized assay (CBASS) make this dataset particularly interesting.

However, I have several concerns about the data analysis and interpretation that should be addressed prior to publication.

The authors emphasize the need for a standardized approach to assess heat tolerance in corals, but do not acknowledge until later in the discussion that CBASS already serves this purpose. Although not named explicitly in the methods, the CBASS framework underlies their experimental design. CBASS is widely used, fast, and effective, with a broad temperature range capable of capturing variation in bleaching response—especially at the threshold, which the authors mention (L98). Moreover, it has been validated in multiple studies (e.g., Cunning et al. 2024; Voolstra et al. 2020, etc.) against longer-term bleaching assays that more closely mimic natural conditions. Given this, the authors must make a much stronger argument for modifying the CBASS approach (e.g., using medians instead of ED50) and for integrating multiple additional metrics (including one that introduces subjectivity).

One of the primary analytical concerns lies in the use of median temperature values to summarize Fv/Fm responses, rather than the standard ED50 metric derived from a dose-response curve. ED50 is biologically meaningful and captures the overall sensitivity pattern across a temperature range. Median values, on the other hand, ignore the curve’s shape (Fig 2) and are more sensitive to the number and selection of temperature points. This may lead to less robust or non-reproducible results and complicates comparisons across studies.

The authors also need to better justify their decision to combine three different metrics (Fv/Fm, pixel intensity, and visual bleaching scores), rather than focusing solely on Fv/Fm, which is the most standardized and quantitative among them. Furthermore, it’s unclear why the authors didn’t simply apply CBASS to the nursery corals (Phase 2), which would have allowed for a quantitative and reproducible ranking without the need to pre-select a single test temperature that would have also accounted for seasonality between phases.

That said, comparing different assessment methods could be very valuable. PAM fluorometers are expensive and can be inaccessible. If visual scores or pixel-based intensity measurements yield rankings that closely match Fv/Fm rankings (e.g., ≥90% agreement), this would be an important and practical finding. But if they do not, the authors need to explain why a composite or alternative approach would be preferable—and what trade-offs are involved. Additional background on the rationale for combining metrics and any previous support for doing so would strengthen the introduction.

The authors write that their goal is to “identify the temperatures that induce the greatest variability in the heat stress responses of seven Caribbean stony coral species.” However, identifying these temperatures would require controlling for factors such as geography, seasonality, and local environment, which aren’t fully addressed here.

Finally, the manuscript would benefit from careful proofreading. There are grammatical issues throughout that detract from the clarity of the results and discussion. In particular, the discussion would be much stronger if it were more concise and clearly focused on the implications of the findings.

Overall, I recommend major revisions to address the analytical concerns, clarify methodological choices, and strengthen the discussion and writing.

INTRODUCTION
The introduction is clearly written and makes a compelling case for the importance of identifying heat-tolerant genets for restoration efforts. The first paragraph effectively sets the stage and highlights the urgency of developing robust methods for assessing thermal tolerance in corals.
However, if the authors intend to prioritize the combination of three metrics (Fv/Fm, pixel intensity, and visual bleaching scores) rather than relying on the CBASS ED50 approach, the introduction should include a clearer rationale for this choice. Specifically, it would strengthen the manuscript to address why CBASS alone—which is increasingly used across the field—may be insufficient, and why these particular metrics were selected as alternatives or complements. This is especially important given that one of the metrics, visual bleaching scores, is inherently subjective.

A brief discussion of the potential advantages and limitations of each metric, and how they might offer complementary insights—or introduce variability—would help frame the rest of the paper and better justify the authors' methodological approach.

MATERIALS AND METHODS
• L159 What was the size range of colonies?
• L163 Please report fragment size and clarify whether fragments were taken from the same colony region (e.g., all branch tips or varied locations).
• L173 Why not call this CBASS methodology?
• L185 Were fragments photographed and visually scored before CBASS? Since the heat pulse doesn't mimic natural bleaching, why was the morning after chosen for visual assessment? Please clarify.
• L188 How many people scored the images? If multiple, how were scorer differences addressed or standardized?
• L186 Can the authors provide a supplemental figure showing example coral fragments for each visual bleaching score to clarify scoring criteria?
• L190 Did the authors assess whether tissue thickness or skeletal porosity influenced visual bleaching scores? These traits could impact visibility of symbiont loss—e.g., scores may be more reliable in thin-tissued or dense-skeleton species than in thick-tissued or porous ones.
• L201 The authors should include a discussion in both the introduction and discussion sections explaining why each of the three methods is weighted equally, especially given their differing levels of subjectivity and accessibility.
• L207-213 This methodology raises concerns about quantitative rigor. With only n=10 per species, subjectivity could influence results. Selecting intermediate temperatures in cases of ties or >50% in a category seems more interpretive than analytical. Can the authors clarify how these decisions were standardized or validated to ensure consistency and reduce bias?
• L220 – Please report the approximate size of fragments and clarify whether a single chunk was collected or if multiple branch tips or sections from different parts of the colony were used.
• L225 – Were all fragments of the same species tested on the same day? If tested over two days, why was this necessary with only two temperatures?
• L238 Can you explain the ranking system in more detail? Were these stress ranges established for all species in the study, or are they more general? Were Fv/Fm scores compared to controls or measured before heat stress to assess if they were already low and did not change? The Evensen paper referenced here noted that visual scores are less precise, but they are low cost, which adds value to comparing them with CBASS ED50 ranking to see which species this method works better for!

RESULTS
• Figure 2 – Using the median may not be the best method here, as the scores don’t seem to directly relate to the curve, and the median is influenced by the number of temperatures selected rather than the temperature at which a 50% drop occurs. I'm curious about why the authors chose not to use the standard CBASS ED50 calculation, which seems to align more naturally with the plots. For example, in ACER, PAST, and OANN, the median appears before we see a drop, which might overemphasize the specific temperatures tested rather than focusing on the actual values.
• L284 Why did the authors choose a model comparing species across all temperatures? Additionally, the current statistical approach doesn't account for the fact that these experiments were conducted at different times of the year. It might be useful to run a model that includes random effects to better test the hypotheses. Additionally, the interpretation of results with phrases like "probably because of..." would be better placed in the discussion section.
• L298 What do the authors mean by "don’t seem to reach"? Can they provide more quantitative details? What is the purpose of the between-species comparison? I understand the within-species comparison, but I'm unclear about the significance of the between-species comparison.
• L299 Please provide p-values when stating significant differences.
• L300 When stating differences, could the authors please include p-values in-line, or alternatively, specify the mean differences with means ± SD directly in the text.
• Fig 2B. What do the authors think PPOR is doing? Seems odd to dip?
• L316 Statistically significant differences? P-value?
• L316 The use of "Interestingly" is a comment on the results, so it should be moved to the discussion. In the discussion, please explain why this is unexpected, particularly regarding the varying ED50 ranges within different species.
• L317 Instead of using "slightly," please provide specific quantitative differences. Similarly, instead of "very close," please report the numeric differences between the values.
• L319 These next sentences belong in methods.
• L321 Why do the authors believe that visual bleaching was not observed? How might this affect the choice of method and its contribution to T50? The lack of correlation mentioned in L327 seems to support this concern.
• L323 Please define "too far from." It would be helpful to use quantitative values throughout the results section for clarity.
• L320 I’m concerned about why and how the authors chose this approach, as it seems to move further away from a quantitative analysis of the data. This decision actually underscores the need for a single, quantitative, repeatable method in the field.
• Fig 4 Consider ordering the data in each figure by thermotolerance score (e.g., for Fv/Fm) rather than colony number, which is arbitrary. This would help the reader more easily see the spread and compare across methods, with all data presented in the same order based on one of your metrics.
• L344 Please define ‘considerable’ and ‘moderate’.
• L346 What does ‘did not seem to impair’ mean? Can the authors report statistics?
• L358 I'm curious about how the fragments went missing. Did they experience complete mortality? Could the authors use the mean of the fragments they still have for the analysis?
• L661 Methods
• L365 Did the authors analyze this by species? It seems like there is a significant species effect driving the PCA.
• L368 Yes, reef likely plays a role. Did the authors perform a PERMANOVA or any statistical tests to account for reef? Also, time of year is an effect we know impacts bleaching and Fv/Fm. How is this accounted for?
• L370 I see the stacked bar plots, but statistical analysis would better support the statement that there is no clear trend.

DISCUSSION
Due to the concerns I’ve raised regarding the data analysis, the discussion will likely change based on revisions.
• L384 The cross-species comparison should note that it does not account for the time of year, which could be an important factor.
• L400 Please provide citation for the OFAV comparison.
• L406 The variation in results across different methods is an important issue in this study. A key aspect of the work is that the authors adapt a widely used methodology (CBASS) and propose combining it with visual assessments. However, the observed variation suggests that this modified approach may have limitations. The paper would benefit from a more in-depth discussion comparing the three methods directly—highlighting their respective strengths, weaknesses, and areas of agreement or discrepancy.
• L415 This is an issue in the experimental design – chosen temperatures were not high enough to measure the response they authors set up to capture, meaning it is challenging to report a T50 or compare methods to assess bleaching if you did not see bleaching.
• L425-27… The current discussion section lacks clarity, particularly in the way contrasting findings are presented. Transitional phrases such as “in contrast” and “on the other hand” are used, but the logical connections between studies are not clearly articulated. It's often unclear what exactly is being contrasted and why. Additionally, this section would benefit from deeper engagement with the cited literature. Rather than listing whether a study found a species to be more or less susceptible to bleaching, could the authors take a more analytical approach.
• L440 (the section) Yes, symbionts specifically play a large role in bleaching phenotype. I’m glad the authors mention this, and am curious that since samples were not taken to identify symbionts, could the authors search the literature more deeply from the area (or nearest location) to identify genus-level differences between species?
• L464 This is the first mention of CBASS, and I’m confused because the experimental design here is a slightly modified version of the CBASS standardized protocol. CBASS was specifically designed to standardize bleaching rank experiments, so the authors would benefit from providing a strong argument as to why their method is preferable to CBASS.
• L469 The method employed in this study does not appear to offer significant advantages over CBASS. In fact, not fully utilizing the CBASS methodology limits the ability to compare this data with the broader body of literature. As the authors note, variability between the three methods and other issues (such as insufficient variation) led to alterations in their own approach.
• L487 Arguably, this shouldn’t matter if the goal of phase 1 was to identify a universal method for the species. However, this finding suggests that using CBASS on the colonies of interest (for a nursery, for example) would be a more reliable approach to ensure success.
• L492 This is an important point. The variation introduced by differences in assessment timing is a significant factor that could impact interpretation of results. This is something the authors could account for in their models by incorporating assessment timing as a variable. Addressing it explicitly would strengthen the analysis and increase confidence. Additionally, this issue supports using a single, standardized assessment per colony rather than a two-phase approach. Streamlining the assessment method could help reduce variability and improve comparability across colonies and studies using CBASS (or one of the other two methods).
• L510 Because genotype assignments are based on quartiles within each experiment, results from future coral samples—even from the same site—won’t be directly comparable. This limits the study’s reproducibility and its value for long-term or cross-study comparisons.
• L532 Tissue thickness and skeletal porosity also influence bleaching responses and should be acknowledged as contributing factors. While studies have shown that CBASS heat tolerance ranking is comparable to longer-term, ecologically relevant heat exposure, we should exercise caution when interpreting changes in coloration in short-term, high-temperature exposures—especially at levels not typically encountered in nature. Additional research is needed to validate whether these short exposures reliably predict bleaching outcomes or rank susceptibility comparably using color.
• L550 The authors should provide a stronger rationale for combining all three assays and weighting them equally, rather than relying on a single, quantitative, and standardized method such as CBASS. While they acknowledge the limitations of their approach, they do not offer compelling justification for why this multi-assay strategy is preferable. An alternative—and potentially more useful—approach would be to compare each method directly to CBASS and offer species-specific recommendations for which alternative method performs best. This would provide clearer guidance for future applications and improve the utility of the findings.

---

## Round 0.2 · accepted · Accept

I thank the authors for drafting one of the most articulate, attentive and thorough "Response to Reviewers" documents I have ever seen. The authors clearly took the time to understand the suggestions of both reviewers and transformed their suggestions into extensive and detailed improvements to the framing and details of the manuscript. They also provided a very detailed response document with extensive direct modifications to the manuscript annotated and addressing every point. The peer review process clearly improved the manuscript and the final product is one that all parties can be proud of.

·

Basic reporting

I found the authors response to reviewers to be thorough and evenhanded. I think the article reads very smoothly and is logically well structured. References are sufficient to justify the assertions made in this study.

Experimental design

I appreciate the more direct discussion of CBASS as a related but alternate approach to that taken by the authors here, which is an improvement over the first round of revisions. I still think there are some issues that the authors didn't fully address around replicability (Reviewer 2 makes a lot of these points in more detail), but I think the approach taken here represents a reasonable trade off between the rigor of CBASS and the specific context and slightly different questions that these authors sought to explore.

Validity of the findings

The findings appear valid, well-structured, and novel. I see no evidence of data manipulation, and the statistical approaches undertaken seem reasonable and account for assumptions appropriately. Where factors are confounding, the authors explain this appropriately and account for them in their choice of analysis. The response to reviewer comments in round 1 helped to clarify the points in the discussion, as well as the paper's overall flow.

Additional comments

Thank you to the authors for your thorough and thoughtful responses.